# MicroGAN: Size-invariant Learning of GAN for Super-Resolution of Microscopic Images

Srishti Gautam, Deepak Kumar Pradhan, Prakash Chandra Chhipa, and
Shinya Nakajima

Arkray AI, Arkray Inc., Japan

**Abstract.** One of the intrinsic problems in deep learning and related
research in microscopic image analysis is the lack of availability of high
quality images due to various factors such as limitations of optics, cost,
etc. Further, there exists extreme variation in sizes of individual cells in
microscopic data which leads to variability in the size of corresponding
images. This demands for size independent inputs to the neural network
which is in contrast to their traditional training procedures. In this pa-
per, we propose MicroGAN, a generative adversarial network (GAN), for
super resolving low quality images without ignoring the original size of
individual cells, thus keeping intact their original texture and anatomy.
Our method consists of a conditional GAN based on trimmed U-Net and
WGAN-GP architecture followed by global average pooling for tackling
the size-invariance problem during the learning phase. We tested our
methodology on urine sediment analysis data which shows a significant
perceptual as well as quantitative improvement. We also show an accu-
racy improvement of 2.37% in the intended classification task with the
super resolved data, thus strongly verifying the proposed approach.

**Keywords:** Super-Resolution · Generative Adversarial Networks · Med-
ical image analysis · Urine Sediment · Deep learning.

## 1 Introduction

High resolution (HR) medical images provide detailed texture and anatomical
information which is crucial for their analysis [1] and are thus the need of the
hour. However, due to various limitations on the grounds of optics, cost, memory
etc [1] [2], there is an extreme lack of availability of the intended resolution data.
The attained low resolution (LR) images may comprise of incomplete and inad-
equate data and thus can lead to an increase in false negatives during diagnosis.

Recently, learning based techniques for super resolution such as [3], [4], which
learn from LR-HR pairs and have recently proven their effectiveness in various
domains, e.g. in geographic, surveillance, scene images etc. Further, with the
exponential increase in the deep learning based research, GANs [5] have boosted
the performance of super-resolved images, [6], [7]. Owing to its huge success,
GAN is also getting some attention in the field of super resolution of medical
images, especially in MRI [1], [2], retinal fundus images [8] and some research in
fluorescence microscopy [9] as well.

| Original images of arbitrary size | 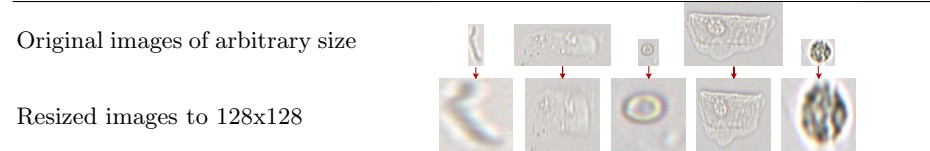 |
|---|---|
| Resized images to 128x128 | |

**Table 1.** Drawbacks of resizing medical images to fixed sizes. The bottom row shows the clear disfigurement of texture, size and shape of the cells.

Nevertheless, the research in GAN based super resolution in microscopic cell images is still at limitation. This can be due to: 1. scarcity of application specific microscopic data, 2. availability of only very deep architectures which focus on more high level semantic features, for example tyre of a car in a street image, and thus not suitable for low level microscopic images consisting of basic texture features, and most importantly, 3. constraint of fixed input-size training, which forces all the images to be pre-processed to a predetermined size, thus losing various important fundamental features in the preliminary stage itself.

To overcome some of the aforementioned drawbacks, our major efforts in this paper are as follows: 1. We propose MicroGAN, a novel GAN architecture, first of its kind in our knowledge designed particularly for size independent super resolution of microscopic cell images. Specifically, we use a trimmed version of the famous U-Net architecture [10] for generator and a simple CNN followed by global average pooling [11] for discriminator to make the learning size independent. We then show that the generated HR images have improved perceptual quality over the traditional fixed input size networks, 2. We demonstrate the importance of using smaller networks for medical images, where we compare the original and trimmed U-Net architectures for the generator along with Deblur-GAN [7]. This further leads to a significant speed up in both training as well as inference times, 3. We confirm our methodology by experimentation on real urine sediment analysis dataset, where we train MicroGAN for generating 35x(NA0.6)-like HR images from 16x(NA0.4) LR images of urine sediments. 4. We compare the final classification results on the raw LR images and the super resolved (SR) images generated by our approach. The results show an increase of 2.37% in the final classification accuracy, thus strongly supporting our methodology.

## 2   Proposed Method

The fundamental idea of GAN defines two main components, a generator G and discriminator D, which compete against each other until they reach a Nash equilibrium. G aims to create as perceptually realistic images as possible to fool D, while D aims to distinguish between the real and fake images.

In the task of super resolution, conditional GANs come into action, which learn an unknown mapping between LR ($I_{LR}$) and HR ($I_{HR}$) images. The objective of super resolution cGAN [6] can be expressed as:

$$\hat{G}, \hat{D} = \min_{G} \max_{D} \; \mathbb{E}_{I_{\mathrm{HR}}} \left[ log\mathrm{D}(I_{\mathrm{HR}}) \right] + \mathbb{E}_{I_{\mathrm{LR}}} \left[ 1 - log\mathrm{D}(\mathrm{G}(I_{\mathrm{LR}})) \right] \qquad (1)$$

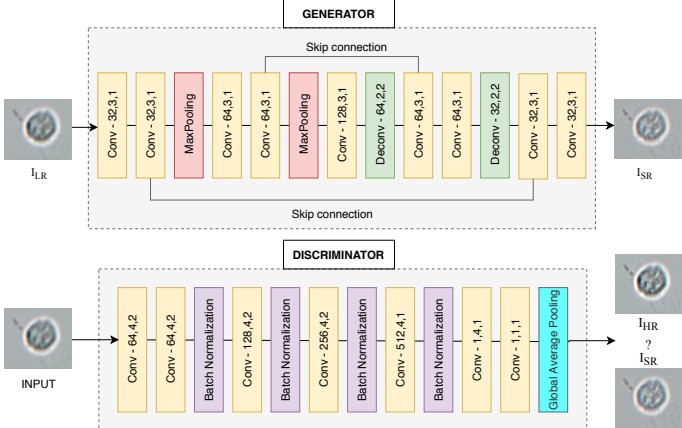

**Fig. 1.** Trimmed U-Net generator and size-invariant discriminator. The numbers mentioned in each block are the number of filters, the filter size and the stride, respectively.

Our aim is to generate HR images from the LR without any availability of model-based priors. For this, we propose MicroGAN, as described in the following sub-sections.

**Generator architecture:** Keeping in mind the emphasis on microscopic cell images, for our generator we use the trimmed version of U-Net architecture. U-Net has been decided upon considering its efficiency in learning the cell image data representation for segmentation [10]. Trimming of the original U-Net is done considering the fact that the deeper layers contribute in recognizing more high level semantic features, which are non-existent in the case of low level single cell images. The detailed generator architecture is shown in Fig. 1. Each layer is followed by ReLU activation, except the last layer which is followed by sigmoid.

**Discriminator architecture:** The discriminator network is identical to the one used in [7], except for the fully connected layers. One major point to be stressed upon here is that although the generator allows any arbitrary sized input, most of the discriminator architectures force the input images to be of same size. In microscopic cell images specifically, there exists extreme variation in the size of images. Resizing these to a fixed size can lead to drastic change in the original texture (Table 1), which can induce false training of models. Therefore, to make the learning size-invariant, we incorporate a very simple yet efficient way, in which the fully connected layers have been replaced by a Global Average Pooling (GAP) [11] layer at the end. This ensures that the training becomes independent of the size of the input images. Each convolutional block is followed by LeakyReLU activation, while the last layer is followed by sigmoid.

**Loss functions:** The loss function used is a combination of adversarial and content losses. For the adversarial loss, $L_G^{SR}$, we use WGAN-GP [12] to prevent

| | BACT | CASTH | CASTO | EC | MCS | RBC | WBC | XTAL | YST |
|---|---|---|---|---|---|---|---|---|---|
| |  | | | | | | | | |
| **Classes** | Bacteria | Cast Hyaline | Cast Other | Epithelial Cell | Mucus | RBC | WBC | Crystal | Yeast |
| **#images** | 826 | 366 | 112 | 952 | 1774 | 987 | 997 | 161 | 634 |

**Table 2.** UrineSed2: sample urine sediment images with total number of images.

vanilla GAN's downsides of unstable training, collapsed mode and difficulties in hyper-parameters tuning [1]. For the content loss, we use perceptual loss, $L_{VGG}^{SR}$ [6], which is based on the difference of the CNN feature maps of original HR and generated SR image. The motivation for using perceptual loss is same as [6], since it is proved to generate perceptually realistic images as opposed to the pixel-wise MSE loss. This was further verified for the case of cell images during our experimentation where perceptually more compelling images were obtained. The final loss is calculated as, $L = L_G^{SR} + \lambda L_{VGG}^{SR}$ ,where we take $\lambda$ as 100 for our experiments, same as in [7].

## 3    Experiments and results

### 3.1    Data preparation

We use urine sediment dataset to evaluate the proposed approach. It is to be noted here that NA (Numerical Aperture) is a measure of the camera's ability to gather light and resolve fine specimen detail. Thus, NA determines the resolving power of an objective. Therefore, the difference in low resolution and high resolution images in this case is in terms of amount of details captured in the same size. We have 135 high resolution multi-cell brightfield microscopic images available, acquired with 35x(NA0.6) objective lens. From these, we extract the single cell images using the following straightforward unsupervised segmentation algorithm: 1. apply adaptive mean thresholding with a window size of 11x11, 2. apply dilation, flood filling and opening morphological operations, 3. finally, crop the single cell images by using bounding box around the segmented regions. Using this, we obtain single cell HR images of varying sizes as shown in Fig. 2. To obtain the corresponding LR images, we apply a gaussian filter with a kernel size of 15x15 on the HR images (Fig 2). This was selected empirically, considering the output's resemblance to original LR images acquired with 16x(NA0.4) objective lens. Thus, we get LR-HR pairs of 16,692 single particle urine sediment images. Out of these, 15,022 images are reserved for training and remaining 1,670 images are reserved for testing (UrineSed1). For testing, we also use completely unseen real LR images acquired with 16x(NA0.4) (UrineSed2, Table 2). The size of all images vary from as small as 30x14 pixels to as large as 765x1607 pixels.

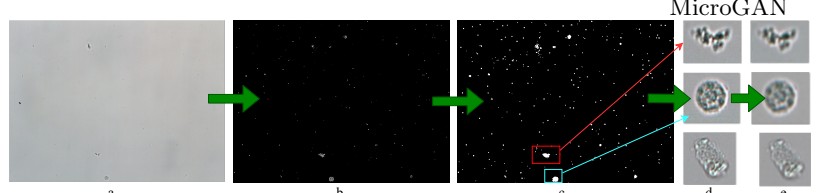

**Fig. 2.** UrineSed1 dataset preparation: (a) Multi-cell HR image of size 4096x3072, (b) Adaptive thresholding, (c) Morphological operations, (d) Segmented $I_{HR}$, (e) $I_{LR}$.

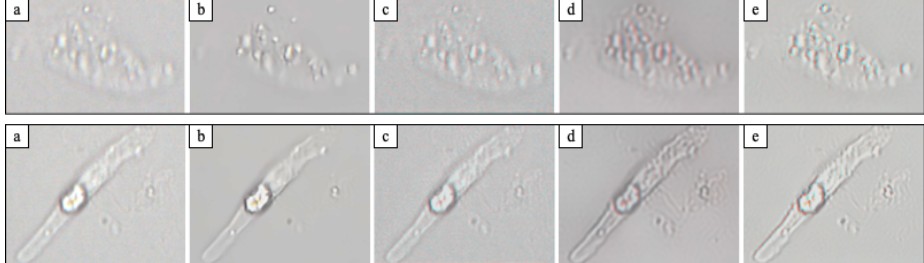

**Fig. 3. Visual results on UrineSed2:** (a) LR image ($I_{LR}$), (b) DeblurGAN [7], (c) MicroGAN-FS, (d) MicroGAN-NT, (e) MicroGAN (proposed).

### 3.2   Training and inference

We trained all of our models using keras deep learning framework with tensorflow as backend. The images are normalized in the range of [-1,1] by subtracting 127.5 and dividing by 127.5 before training. For optimization, Adam is used with $\beta = 0.9$ and learning rate of $lr = 10^{-4}$. Since we employ size-invariant learning, on-line training is used where the network is trained for each example for a total of 200 epochs. In each epoch, to develop a strong discriminator, the critic network is trained 4 times more than the generator. For inference, a single LR image is passed once through the generator network to obtain the corresponding SR image. Since the generator can take any arbitrary sized input, images are not resized at the inference time for any of the experiments.

### 3.3   Size-invariant discriminator analysis

To demonstrate the effectiveness of the proposed size-invariant GAN, we compare its results with the traditional fixed input size GAN (MicroGAN-FS). For this, we keep the architecture of generator same as MicroGAN. For the discriminator network, as an alternative to the last layer of global average pooling, we implement a fully connected layer with 1024 neurons and ReLU activation followed by the last layer with 1 neuron with sigmoid. Considering these architectural changes, since we now need fixed-size input for the training of MicroGAN-FS, we resize all images to 128x128. The visual results in Fig. 3 and Fig. 4 show better results by MicroGAN for super resolution of both UrineSed1 and UrineSed2 images. We note here that due to non availability of HR images for UrineSed2, these are not shown in Fig. 3. MicroGAN-FS lacks in enhancing the some of the prominent details when compared to MicroGAN, which is clearly visible in Fig. 4 due to the loss of various characteristic features during resizing.

Table 3 shows the comparison of mean PSNR and SSIM values along with number of parameters to be learned followed by mean inference time for 1670 test images from UrineSed1. Quantitatively, in terms of SSIM, PSNR, Micro-GAN performs better than MicroGAN-FS. This stresses upon the efficiency and importance of the size-invariant learning in medical images. Additionally, in-spite of having a very easy task of reconstructing Gaussian blurred images while training, the same network performs exceptionally well on unseen real data as well (UrineSed2), which further confirms the efficiency of the proposed method. Additionally, it can be seen in Table 3 that the inference time of MicroGAN-FS lower than MicroGAN. This is because all the images have been resized to 128x128 for MicroGAN-FS but for MicroGAN, original images are used which are varying from 30x14 to 765x1607 pixels, which leads to a slight variation in the average inference times.

### 3.4   Trimmed U-Net generator analysis

In this work, we also emphasize on the importance of using smaller architectures for medical images. For this, we compare the results of MicroGAN with a similar state-of-the-art architecture (DeblurGAN [7]), which has originally been developed for motion deblurring. It consists of a very deep generator network with 11 million parameters. We further compare the results of MicroGAN with an alternative architecture comprising of original U-Net as generator (MicroGAN-NT: full U-Net, with no trimming) and the discriminator same as that of MicroGAN. As can be seen clearly in Fig 3 and 4, MicroGAN generates better perceptual results than both DeblurGAN and MicroGAN-NT. This is because the microscopic cell images are made up of low level features which define the texture of the cells. Thus, for these types of images, because of absence of higher level semantic features, the important textural features get lost in the deeper layers of architecture. As can be observed in Fig. 4, MicroGAN-NT enhances the high level edge information and fails to capture intricate texture patterns. DeblurGAN is also unable to define the smaller texture details, which are better captured by MicroGAN. Additionally, both MicroGAN-NT and DeblurGAN consist of very high number of parameters and hence run very slow at inference time too. Table 3 shows better performance of MicroGAN in terms of PSNR and SSIM than MicroGAN-NT and DeblurGAN. This shows that MicroGAN outperforms in capturing both the prominent edge information as well as the fine grained texture details and thus validates the importance of using smaller architectures for medical images.

|          | DeblurGAN | MicroGAN-FS | MicroGAN-NT | MicroGAN |
|----------|-----------|-------------|-------------|----------|
| PSNR(dB) | 27.29     | 29.92       | 25.25       | **30.75** |
| SSIM     | **0.81**  | 0.80        | 0.79        | **0.81** |
| #Param   | 11.401M   | 3.132M      | 19.974M     | **3.063M** |
| Time(s)  | 0.052     | **0.021**   | 0.064       | 0.022    |

**Table 3.** Quantitative comparison for different architectures for UrineSed1.

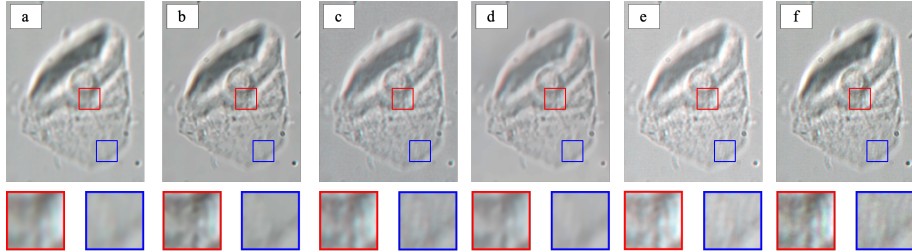

**Fig. 4. Visual results on UrineSed1:** (a) LR image ($I_{LR}$), (b) DeblurGAN [7], (c) MicroGAN-FS, (d) MicroGAN-NT, (e) MicroGAN (proposed) (f) HR image ($I_{HR}$).

### 3.5    Urine sediment classification

Finally, to substantiate MicroGAN's further impact, we compare classification results on the original and transformed SR data from MicroGAN as well as DeblurGAN for urine sediment analysis [13] [14]. The detailed description for the dataset used for this experiment (UrineSed2) is given in Table 2. We fine tune ImageNet pre-trained VGG16 for the classification on all three datasets. We further state that to make fair comparison among the datasets depending only on their pre-processing and independent of classifier, we resize them to 224x224 for classification. For fully connected layers, we have used a hidden layer of 512 neurons followed by final output layer of 9 neurons. The training parameters are identical for all of the experiments ($lr = 10^{-5}$, Adam optimizer with $\beta = 0.8$, Number of epochs = 20). We have a total of 6820 images out of which we use 80% data for training, 10% for validation and remaining 10% for testing and use 5-fold cross validation. The normalized confusion matrix for the test data of original and MicroGAN SR data classification are shown in Fig. 5. This shows an improvement in the accuracies of most of the classes. Additionally we want to state that for urine sediment analysis the most struggling classification is Cast Hyaline vs Mucus. As can be seen clearly in Fig. 5, the CASTH data is also classified with a better accuracy than the original data, owing to a good super resolution of particles in CASTH images by MicroGAN (Fig. 3). The false positive ratio of CASTH to MCS is reduced from 0.25% to 0.11% by using MicroGAN. We also acknowledge that more of the WBC's are misclassified with EC using MicroGAN, which might be because of the increased similarity of WBC cells to rounded EC cells after super resolution. The average training, testing and validation accuracies alomng from 5-folds for original data, SR data by DeblurGAN and SR data by MicroGAN are shown in Table 4. The high resolution accuracies are not reported here because of unavailability of HR data for UrineSed2. It can be seen clearly that there is an overall improvement by using MicroGAN data with an average increase of 2.37% in the test accuracy.

|  | Training Accuracy | Validation Accuracy | Testing Accuracy | Testing F1 score |
|---|---|---|---|---|
| Original | 90.02% | 88.77% | 90.98% | 0.77 |
| DeblurGAN | 92.53% | 89.81% | 92.02% | 0.83 |
| MicroGAN | **94.11%** | **91.88%** | **93.35%** | **0.85** |

**Table 4.** 9-class classification results on UrineSed2 dataset. The values reported here are an average of 5-fold cross validation.

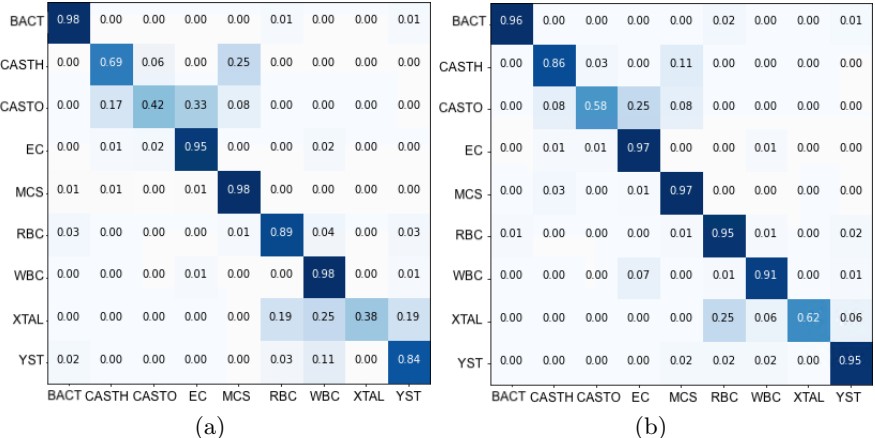

(a)                                                (b)

**Fig. 5.** Normalized confusion matrices for (a) Original Data & (b) SR data by Micro-GAN. Rows depict the actual class and columns depict the predicted class.

## 4  Conclusion

We have described MicroGAN, a GAN architecture for super resolution of size independent microscopic images. The proposed MicroGAN shows qualitative as well as quantitative improvement in terms of PSNR, SSIM, inference time as well as classification accuracy over a state-of-the-art (DeblurGAN) and other alternative architectures on urine sediment dataset. The proposed method can be extended to other medical image domains to achieve super resolution and other related machine vision tasks, such as segmentation, classification, etc.

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
