# OpenReview forum: "MicroGAN: Size-invariant Learning of GAN for Super-Resolution of Microscopic Images"
_MICCAI.org/2019/Workshop/COMPAY — COMPAY 2019_

### Official Review · AnonReviewer2 · 2019-07-26
**MicroGAN: size-invariant learning of GAN for super-resolution of microscopic images.**

**Rating:** 6
**Confidence:** 5

**Review:**

Summary:
The authors trained a GAN-based method to perform super-resolution image reconstruction on images with varying input size. They achieve so by training a UNet-based generator to transform low resolution into high resolution images, then feeding these images into a discriminator that discerns whether they look realistic or not. Crucially, the discriminator can deal with arbitrary input size images, bypassing artifacts caused by rescaling to a fixed input size. Furthermore, they show that downstream classification performed on the proposed reconstructed images is superior to using other super-resolution methods.

Strong points:
* The paper is clearly written, with a good motivation and well-explained problem statement.
* The choice of methodology seems appropriate to solve this problem.

Constructive criticism:
* Methodological novelty is limited to using global average pooling in the discriminator. By doing so, the authors can train with arbitrary input size images.
* In section 3.5, the authors describe the network architecture used for the downstream classification task. They mention that it is a pretrained VGG16, followed by fully connected layers. How did they cope with input size variance? I assume they used global pooling before the fully connected layers.
* In the same section, they report an average accuracy score. I wonder how this accuracy was computed. Does it take into account class imbalance?
* Regarding the same metric (Table 4), it would be preferable to repeat the training and evaluation procedure a few times, reporting confidence intervals or mean+std metrics, in order to show whether the performance difference is real or caused by random initialization.
* Table 4 could also contain a column with performances obtained using the high-resolution images, in order to provide an upper bound in performance for reference purposes.
* Figure 3 could include high-resolution images for reference purposes.
* Additionally, it would be interesting to visualize images that were misclassified with the “original” (low res) network, and correctly classified with the “microgan” approach. This would give an idea of the visual features created by the GAN that pose an advantage towards classification performance.
* The authors should be more cautious with statements like “high-level semantic features” since it is not clearly defined whether edge information is high or low, same with texture.
* Similarly, I do not see any evidence suggesting that “smaller architectures are more suitable for medical imaging”. In any case, smaller architectures are suitable for smaller training sets, because of their regularization effect (and medical imaging datasets tend to be small).
* The sentence “Authors Suppressed Due to Excessive Length” appears several times along the paper. It would look better if the authors’ names would show up instead.

---

### Official Review · AnonReviewer3 · 2019-07-26
**Review of paper #28**

**Rating:** 6
**Confidence:** 3

**Review:**

Summary : the paper proposes an approach for the super-resolution of microscopic images.

The paper is very well written and easy to follow. The motivation of the approach is clearly described, the proposed approach well introduced with respect to the actual state-of-the-art and the experimental results are good.

So this is basically a good paper. However the level of novelty is low.
The proposed principle is to :
- combine the approaches of [10] and [7] for the Generator/Discriminator structures
- slightly modify [7] to be image size-invariant
- combine two common losses [12, 6] for minimization

The modification of [7] to be size invariant is not well motivated. If the problem relies on having a FC layer at the end of the network, then a bottleneck layer would also do the job. More motivation is needed here, especially since this is the main novelty of the paper.

The paper is dedicated to the super-resolution of microscopic images but :
- It is not clear why the combination of a wasserstein loss and a perceptual loss that have been designed for natural images can be interesting and accurate for microscopic images
- the kind of microscopic images that is considered in the paper is very restrictive : as fas as I can see, these are unstained images and these images have very specific visual properties that are very different from other microscopic images (e.g., histological ones). I think the results would benefit from also studying another type of microscopic images.

Usually a LR images has a lower resolution than the HR image, this does not seem to be the case here. Please elaborate more on that.

Why is the time of MicroGAN-FS lower than MicroGan in Table 3 whereas the latter has a simple architecture (GAP vs FC) ?

In Section 3.5, please provide also the F1-score.

---

### Official Review · AnonReviewer1 · 2019-08-08
**MicroGAN: Size-invariant Learning of GAN for Super-Resolution of Microscopic Images**

**Rating:** 7
**Confidence:** 3

**Review:**

The paper proposes a GAN for classification of cell images of varying sizes. The method is based on a number of previously proposed approaches and includes a proper comparison to those and others. The evaluation is on a relatively large set.

Adapting networks to deal with input of various sizes is a worthwhile topic and the authors' motivation for their methodological choices is clear. I am not completely convinced by the first results, however. I personally do not see a  clear improvement of the proposed method over DeblurGAN (figs 3&4). Moreover, the results in figure 5 vary. My conclusion would be that the methods performance similarly, but the proposed approach has the advantage of using a smaller network.  I do disagree with the authors' claim that smaller networks are important for medical images. It is not substantiated by the experiments.

Nonetheless, the aim is certainly of interest. Perhaps the current problem is not suitable to demonstrate the performance differences well. The classification problem does not appear to be very hard, giving the substantial differences between the classes (table 2). I would advise the authors to focus on another, more challenging problem as well.

---

### Decision · Program_Chairs · 2019-08-20

Accept